# Differential Expression of *HOX* Genes in Mesenchymal Stem Cells from Osteoarthritic Patients Is Independent of Their Promoter Methylation

**DOI:** 10.3390/cells7120244

**Published:** 2018-12-05

**Authors:** Arkaitz Mucientes, Eva Herranz, Enrique Moro, Cristina Lajas, Gloria Candelas, Benjamín Fernández-Gutiérrez, José Ramón Lamas

**Affiliations:** 1Instituto de Investigación Sanitaria del Hospital Clínico San Carlos (IdISSC). UGC de Reumatología, Hospital Clínico San Carlos, 28040 Madrid, Spain; arkaitzmucientes@gmail.com (A.M.); evaherranzdlp@hotmail.com (E.H.); cristinajesus.lajas@salud.madrid.org (C.L.); gloria.candelas@salud.madrid.org (G.C.); benjamin.fernandez@salud.madrid.org (B.F.-G.); 2Instituto de Investigación Sanitaria del Hospital Clínico San Carlos (IdISSC). UGC de Traumatología, Hospital Clínico San Carlos, 28040 Madrid, Spain; enrmoro@hotmail.com

**Keywords:** osteoarthritis (OA), cartilage, bone, *HOX*, homeobox, mesenchymal stem cells, epigenetics, methylation

## Abstract

Skeletogenesis, remodeling, and maintenance in adult tissues are regulated by sequential activation of genes coding for specific transcription factors. The conserved Homeobox genes (*HOX*, in humans) are involved in several skeletal pathologies. Osteoarthritis (OA) is characterized by homeostatic alterations of cartilage and bone synthesis, resulting in cartilage destruction and increased bone formation. We postulate that alterations in *HOX* expression in Mesenchymal Stem cells (MSCs) are likely one of the causes explaining the homeostatic alterations in OA and that this altered expression could be the result of epigenetic regulation. The expression of *HOX* genes in osteoarthritic-derived MSCs was screened using PCR arrays. Epigenetic regulation of *HOX* was analyzed measuring the degree of DNA methylation in their promoters. We demonstrate the downregulated expression of *HOXA9* and *HOXC8* in OA-MSCs. However, their expression does not correlate with promoter methylation status, suggesting that other epigenetic mechanisms could be implicated in the regulation of *HOX* expression. Studies on the role of these genes under active differentiation conditions need to be addressed for a better knowledge of the mechanisms regulating the expression of *HOX*, to allow a better understanding of OA pathology and to define possible biomarkers for therapeutic treatment.

## 1. Introduction

Osteoarthritis (OA) is characterized by synovial joint cartilage deterioration, inflammation, and osteophyte formation. It is the most prevalent and disabling degenerative joint disease in the elderly [1]. The exact pathogenetic mechanisms of OA are unknown. However, the progression and severity of OA are attributed to environmental and genetic factors [2,3,4]. Two features of endochondral ossification, chondrocyte hypertrophy and apoptosis, have been proposed as mechanisms of OA initiation [5]. Physically and biochemically, cartilage and subchondral bone are closely related structures of the joint affected in OA. Their implication likely results from alterations of their molecular crosstalk, leading to the imbalance between new tissue formation and degradation, preventing proper self-repair [6,7]. Additionally, the key role of inflammation in OA progression is well documented [8,9].

At the moment, OA remains incurable, and nonsurgical treatments are focused on symptom management and pain relief [10]. Other treatments based on the inhibition of inflammation or stimulation of cartilage formation slow down OA progression, but are unsatisfactory to maintain joint function in the long term [11].

The need for new therapeutic options has focused on the possibility of regenerative medicine as one of the most promising treatments for OA [12,13,14]. Mesenchymal stem cells (MSCs) can differentiate into several cells, including chondrocytes and osteoblasts, which in turn determine the structure and specific properties of joint tissues. Although joint development occurs early in the embryonic stages, the regeneration and renewal of adult tissues are driven by similar biological mechanisms [15]. During endochondral skeletal development, the cellular condensation of mesenchymal cells form an aggregate of pre-chondrocytes [16]. Once chondrocytes mature and proliferate, they secrete extracellular matrix (ECM), undergo hypertrophy, and die. The embryonic skeleton, initially formed by an avascular hyaline cartilage template, is further colonized by blood vessels carrying progenitor bone-forming cells. 

The Homeobox genes (*HOX*, in humans) are among the most important genes controlling morphogenesis and embryonic skeletal formation through endochondral ossification, but also have a role in skeletal regeneration [15,17,18]. *HOX* genes encode a group of transcription factors involved in the regulation of multiple functions, including differentiation and development of stem cells. These genes, characterized by a highly conserved region of 180 base pairs of DNA called the homeobox, code for a 60-amino acid helix–turn–helix DNA-binding domain, known as the homeodomain. According to their genomic location, homeobox genes can be roughly described as clustered *HOX* genes in the strictest sense and non-clustered ones. Although their exact role in cellular systems and in adult tissues is still under investigation, they share similar known functions. (Appendix A). In humans, the 39 existing *HOX* genes distribute into four groups: *HOXA*, *HOXB*, *HOXC*, and *HOXD*, originated by genomic duplications and mapped at four different chromosomal locations. Each group contains different genes, with sequence homology within groups, known as paralogous genes (Appendix A).

Because the MSCs are progenitor cells, defects or alterations in their differentiation program in response to the cell environment or an evolutive program could explain the downstream homeostatic alterations present in OA. According to their regulatory function, it is more than reasonable to recognize the key role of *HOX* genes in different pathologies, e.g., hematological malignancies [19] and other conditions characterized by metabolic and skeletal dysfunctions (Appendix A) [20]. However, their role in OA pathogenesis has been little studied. Their involvement in early phases and during the development of OA has been suggested [21]. In mice, it has been demonstrated that several *HOX* genes are responsible for controlling the osteochondrogenesis, longitudinal growth, and maturation of skeletal structures [22]. Moreover, the role of *HOX* in OA was suspected by our group after data mining of a transcriptome dataset previously published, where we observed a differential regulation of *HOX* genes in OA and non-OA MSCs [23].

Given their essential functions, *HOX* protein expression are subjected to complex regulatory mechanisms. Epigenetic regulation of *HOX* occurs by any of the common described mechanisms, including miRNA regulation [24], chromatin modifications on histones, or DNA methylation. The role of methylation in osteoarthritis has been studied in epigenome-wide studies, revealing the implication of this mechanism in several inflammatory factors, as well as in several *HOX* genes [25].

The role of *HOX* genes in adult regeneration processes is still an unexplored area of research of great importance in the context of regenerative therapies. In this work, we studied the expression of *HOX* genes in MSCs and their correlation with their promoter methylation status. A better knowledge of these mechanisms regulating the expression of *HOX* genes is essential both to improve the understanding of OA pathology and to define possible biomarkers for the detection or specific therapeutic treatment of this pathology.

## 2. Materials and Methods

### 2.1. Array Data Analysis

The initial hypothesis of this work was generated after data mining of data from a DNA array (Agilent “Human Genome, Whole” annotation data; chip hgug4112a), published by our group [23]. Briefly, after microarray scanning, numerical data were processed using the Agilent Feature Extraction image analysis software Version 9.1.3.1 (AFE) (Agilent, Santa Clara, CA, USA). The data were analyzed in R (R Development Core Team) using packages of the Bioconductor project20 (Buffalo, NY, USA) as well as custom written R routines.

### 2.2. Patients

Bone marrow samples were obtained from the femoral channel at the time of surgery for total hip replacement of six patients with osteoarthritis (five males and one female. Median age 71 years, range 59–75) and six control donors suffering traumatic sub-capital fracture (three males and three females. Median age 79 years, range 67–91). OA diagnosis was established according to the American College of Rheumatology criteria. Control donors did not show radiographic changes of OA or osteoporosis (densitometric T-score > −2.5 SD). Written informed consent was obtained from all patients before sample collection. The study was approved following the guidelines of the institutional ethics committee (Comité Ético de Investigacion Clínica Hospital Clinico San Carlos—Madrid, Spain) and the principles expressed in the Declaration of Helsinki.

### 2.3. Cell Cultures

Human MSCs were obtained from bone marrow aspirates of patients undergoing surgery for total hip replacement. Bone marrow aspirates were diluted in an equal volume of saline and centrifuged over a Ficoll layer at 2000× *g* for 20 min. The cellular fraction was recovered and washed two times in Dulbecco’s Modified Eagles Medium (DMEM) (Lonza). The cell pellet was suspended in 5 mL with culture medium (DMEM supplemented with 2 mM glutamine, 0.06% penicillin, 0.02% streptomycin, and 10% FBS). The cells were cultured in a 5% CO_2_ humid atmosphere at 37 °C in 25 cm^2^ flasks. Confluent monolayers of cells were obtained, refreshing the medium every two days and removing nonadherent cells. The cells were passaged two more times until confluent, then cryopreserved for later studies.

### 2.4. Cell Characterization

#### 2.4.1. Flow Cytometry

Immunophenotyping was performed on cell cultures recovered from cryopreserved stocks at the third passage. The cells were grown to confluence, and expression markers were evaluated by flow cytometry using the following antibodies: Mouse antihuman IgG1 antibodies: CD73, CD90, and CD105 as positive expression markers, and CD14, CD34, and CD45 (rat antihuman IgG2b) as negative expression markers. All the antibodies and isotype controls were R-phycoerytrin (PE)-conjugated (Miltenyi Biotech, Bergisch Gladbach, Germany). Immunostaining was performed incubating for 30 min at 4 °C. After washing, the cells were fixed with 0.1% paraformaldehyde prior to analysis (Appendix A).

#### 2.4.2. Histochemistry

Cells from cryopreserved stocks were grown to confluence in 12-well plates and subjected to osteogenic and chondrogenic differentiation using the appropriate induction media (Lonza osteogenic medium Cat# PT-4120, and chondrogenic Cat# PT-3003) supplemented with 10 ng/mL TGFβ3 Cat# PT-4124 under conditions described by the manufacturer.

The multilineage differentiation potential of MSCs was assessed by histochemical staining and phase-contrast microscopy (Leica 4000b DMI, Leica Microsystems GmbH., Wetzlar, Germany). The degree of mineralization in osteogenic cultures was assessed by staining with 2% Alizarin Red S (Sigma-Aldrich, Saint Louis, MO, USA). Briefly, the cells were washed three times with phosphate buffered saline (PBS) pH 4.2 and fixed 20 min with 4% paraformaldehyde. The fixed cells were washed, stained, and washed again to remove excess stain. Similarly, the chondrogenic potential was evaluated by measuring the production of proteoglycans produced by chondrocytes after staining for 20 min with 1% Alcian blue (Sigma-Aldrich, Saint Louis, MO USA) (Appendix A).

### 2.5. Determination of Human HOX Gene Expression by RT-PCR

RNA was isolated from RNAlater conserved samples using the SPEEDTOOLS Total RNA Extraction kit (ThermoFisher, Waltham, MA, USA). RNA was further retrotranscribed using the Maxima H Minus Reverse Transcriptase (SABiosciences, Bydgoszcz, Poland), following the manufacturer’s instructions. The expression of 38 human *HOX* genes and three Endogenous Control Genes was analyzed using the TaqMan^®^ Gene Expression Assays, using primers labeled with FAM™ (Appendix A) and TaqMan^®^ Universal PCR Master Mix. RT PCR was performed in a Mastercycler realplex4 epgradient S (Eppendorf, Hamburg, Germany) using the following conditions: 2 min at 50 °C, 10 min at 95 °C, and 40 cycles of melting (95 °C for 15 s) and annealing/extension (60 °C for 60 s). Baseline and threshold values of amplification plots were set automatically by the instrument.

The relative gene expression for each gene was calculated using the comparative 2^−ΔΔCt^ method. ΔCt for each sample was normalized using the average of gene Ct values proven to be the most stable across samples. Detectable PCR products were obtained, and Ct values > 35 cycles were considered nonspecific and discarded for further calculations. The raw data were then analyzed to calculate the fold change values expressed as 2^−ΔΔCt^ for genes in OA-MSCs relative to Control-MSCs.

### 2.6. Measurement of HOX Genes Promoter Methylation

To determine the degree of methylation in the promoter region of the genes of interest, we performed a methylation-sensitive restriction qPCR analysis of 96 genes, using the cataloged EpiTect Methyl II PCR assay for Human Homeobox EAHS-3560Z (SaBiosciences/Qiagen, Venlo, Netherlands). PCRs were run in a LightCycler^®^ 480 (Roche, Basel, Switzerland). Briefly, the method was based on the detection by real-time PCR of the remaining input DNA after enzymatic cleavage with methylation-sensitive and methylation-dependent restriction enzymes. The method uses primers flanking the restriction sites of a CpG region of interest. Digested DNA was used as a template for qPCR Assay using the RT² qPCR SYBR Green/ROX MasterMix (Qiagen, #330523, Venlo, Netherlands) under standard amplification conditions described in the product specifications protocol.

### 2.7. Statistical Analysis

Statistical analysis was performed using One-way ANOVA corrected for multiple comparisons, followed by the Bonferroni’s post hoc test or Student’s *t*-test (unpaired; two-tailed) with a significance of *p* < 0.05 (Prism Graph Pad Software, San Diego, CA, USA). Microarray data statistics were as described in Reference [23]. Gene sets multiple comparisons were considered significant for a False Discovery Rate below 25% (FDR adjusted *p*-values or *q*-value < 0.25).

## 3. Results

### 3.1. Microarray-Based Gene Expression Analysis Reveals the Lower Expression of HOX Genes in OA-MSCs

The rationale for this study arises after the analysis of differential expression data, obtained by our group, using a DNA array (Agilent “Human Genome, Whole” annotation data; chip hgug4112a). The expression profiles of bone marrow-(BM) MSCs from eight paired OA patients and patients with hip fracture without OA signs were compared. The analysis revealed a significant differential expression of 334 genes between Control-MSCs and OA-MSCs. Of these, in OA-MSCs, 152 genes were at least two times upregulated, and 182 were at least two times downregulated. Interestingly, the supervised examination of these data revealed that *HOXC13*, *HOXB6*, *HOXB4*, and *HOXB3* were among the top 100 OA downregulated genes (Appendix A). In order to determine the statistical relevance of these genes, several gene sets were designed and analyzed between the two phenotypes using a computational Gene Set Enrichment Analysis (GSEA) by means of the available tool at http://www.broadinstitute.org/gsea [26].

The existence of significant and concordant differences between OA and control samples in a gene set containing the 39 human *HOX* genes was analyzed. The results showed the existence of a significant upregulation of the *HOX* gene set in the control phenotype (at nominal *p*-value < 1% (0.0037) and FDR *q*-value < 0.25). The genes contributing to this enrichment were *HOXA2, HOXA4, HOXB2, HOXB3, HOXB4, HOXB6, HOXB13, HOXC8, HOXC12*, and *HOXC13*. The results are depicted in Figure 1. Moreover, when *HOX* genes were partially analyzed using one gene set per cluster, the results revealed the upregulation of *HOXA* and *HOXD* clusters in OA phenotype, being *HOXA* the only significantly enriched gene (FDR *q*-value = 0.003). Similarly, in control samples, two clusters were also upregulated, being the *HOX*B gene set the one significantly enriched (FDR *q*-value = 0.2016) (Appendix A).

### 3.2. Identification of HOX Gene Biological Function Signatures Enriched in MSCs

Once this preliminary gene set enrichment analysis (GSEA) was performed, the next goal was to determine the contribution of gene subsets created according to the biological functions in which the *HOX* gene participate. The OA phenotype (eight samples) revealed that five out of 12 gene sets were upregulated in the OA phenotype. None of the upregulated sets were significantly enriched at nominal *p*-values < 1% or < 5%, but one gene set was significant at FDR < 25%. Similarly, seven out of 12 gene sets were upregulated in the Control phenotype. One gene set was significantly enriched at nominal *p*-value < 1%, and two gene sets were enriched at *p*-values < 5%. Five gene sets were significantly enriched at FDR < 25% (Appendix A).

### 3.3. Determination of Specific HOX Expression Pattern by RT-PCR

To validate the differentially expressed genes identified by expression arrays, a reverse transcription-polymerase chain reaction (RT-PCR), specific for 38 out of the 39 *HOX* genes, was carried out. This analysis included new BM-MSCs, different from those used in the previous expression arrays. A total of six OA-MSCs and six Control-MSCs were analyzed. The expression differences found were not greater than 2.9 times. After multiple test corrections, only *HOXA9* and *HOXC8* downregulated expression in OA MSCs was significantly different (Figure 2 and Appendix A).

### 3.4. Epigenetic Methylation Status of HOX Gene Promoters

Progressive chromatin “exposure” is likely one of the main causes of *HOX* sequential expression and temporospatial activation. Although chromatin accessibility greatly depends on histone modifications, the methylation status of gene promoters can also influence their transcriptional expression. Our aim was to evaluate the existence of any particular methylation pattern of *HOX* genes able to explain their differential expression between OA-MSCs and Control-MSCs.

Thus, we performed a methylation-sensitive restriction qPCR analysis of 96 genes included in the EpiTect Methyl II PCR assay (see Materials and Methods). Unfortunately, this panel only include 34 out of the 39 *HOX* genes. Our results indicate that *HOXA2, HOXC10*, and *HOXC12* promoters were hypermethylated in OA-MSCs, while the *HOXA11, HOXB5, HOXB6, HOXB8, HOXD3, HOXD8*, and *HOXD11* promoters were hypermethylated in Control-MSCs. Graphical results are depicted in Figure 3.

## 4. Discussion

Homeobox genes are well-known regulators of proliferation and differentiation processes during embryonic development. Moreover, they participate in adult tissues beyond embryogenic stages [15]. Although they are not skeletogenic-specific factors, *HOX* expression is present in MSCs as well as in adult tissues and organs. Their importance is of particular interest in biological processes where cell proliferation, differentiation, and maturation are implicated. Genetic loss-of-function studies have provided evidence that *HOX* altered expression is present in many diseases; its role in the development of tumor processes is well known. Since *HOX* are transcription factors, their target genes can be the biological effectors governing skeletal patterning during endochondral ossification as well as repair and regeneration in the adult skeleton.

Alone or combined with carriers or scaffolds, the use of MSCs is being actively explored for tissue engineering, particularly in regard to cartilage and bone repair in OA [27]. Chronic inflammatory diseases can induce in MSCs early signs of senescence, which may contribute to the pathogenesis of the disease as a result of defective MSCs self-renewal and cell fate determination [28]. Moreover, a gene expression profile has also been associated with the abnormality of BM-MSCs [29]. Here, we screened the expression of human *HOX* in MSCs to determine their possible involvement in OA pathogenesis.

Our data show that 12% and 5% of the 39 *HOX* genes were downregulated and upregulated, respectively, in MSCs from OA individuals. Of these, only *HOXA9* and *HOXC8* among the downregulated genes showed statistical relevance. Interestingly, in a murine model of embryonic fibroblasts, Lei et al. identified changes in the expression of 34 genes after *HOXC8* overexpression. Further chromatin immunoprecipitation analysis (ChIP) revealed the direct interaction between *HOXC8* and the osteopontin promoter (OPN) and, subsequently, reduced OPN expression, confirming that OPN is a *HOXC8* direct target [30]. Similarly, other authors have shown that *HOXA9* overexpression is able to induce the expression of genes associated with the establishment of typical stemness characteristics, such as invasive potential, migration, etc., properties that can be reverted after *HOXA9* silencing [31]. Collectively, these pieces of evidence seem to indicate that OA-MSCs have a partial loss of their stemness and simultaneously adopt a phenotype with typical characteristics of bone tissue.

There is evidence that the regulation of gene expression, and in particular the collinear expression of *HOX* genes, is caused by different mechanisms, including the progressive “exposure” of chromatin and alterations in their three-dimensional conformation induced by epigenetic mechanisms, such as DNA methylation and post-translational modification of histones [32,33].

In order to determine whether the expression of *HOX* genes is determined by a differential methylation pattern of their promoters, we conducted a study of the methylation status of these genes. 

Although methylation differences in the promoter could be expected to correspond to a lower gene expression, this is not the case in our study. Although we arbitrarily chose a differential threshold of 10%, it does not seem, at least in our case, that the mechanism of gene regulation is the promoter’s methylation, since even the two *HOX* genes most expressed have similar methylation levels, both in in OA-MSCs and in Control-MSCs (Figure 3). 

## 5. Conclusions

In general, the differential degree of methylation of *HOX* gene promoters, between OA and Control-MSCs, is greater and does not correlate with the number of genes differentially expressed. Particularly, the downregulation in OA-MSCs of *HOXC8* and *HOXA9* expression, as well as the upregulation of *HOXA13* and *HOXC13* occur despite the similar promoter methylation. This could be explained as the result of MSC phenotype stabilization in vitro in the absence of differentiation, migration, or other active cell functions [34]. Thus, further studies of *HOX* function are needed, including analyses of their sequential expression, activation, and functionality under active differentiation conditions to better determine the pathophysiological role of methylation and other *HOX* epigenetic regulatory mechanisms or the involvement of other epigenetic mechanisms not covered in this manuscript.

## Figures and Tables

**Figure 1 cells-07-00244-f001:**
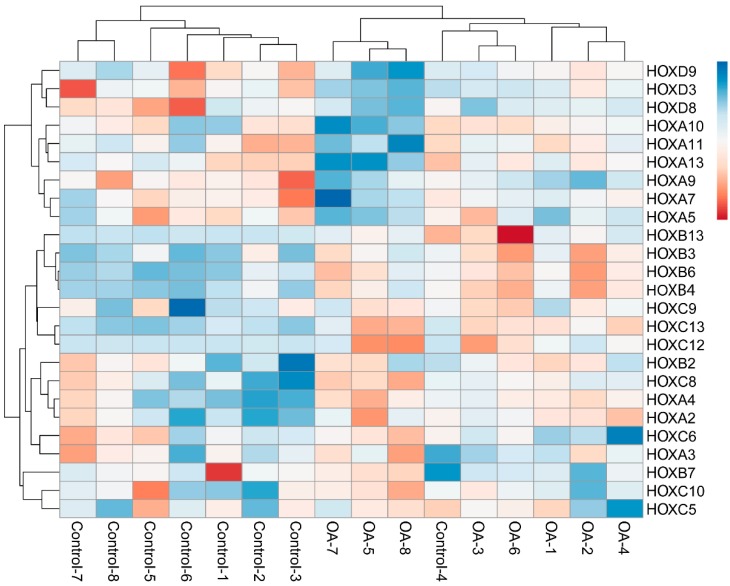
Heat map and clustering of Homeobox gene (*HOX*, in humans) expression for osteoarthritis (OA) and Control phenotypes. The upregulated and downregulated genes are shown in bluish and reddish colors, respectively.

**Figure 2 cells-07-00244-f002:**
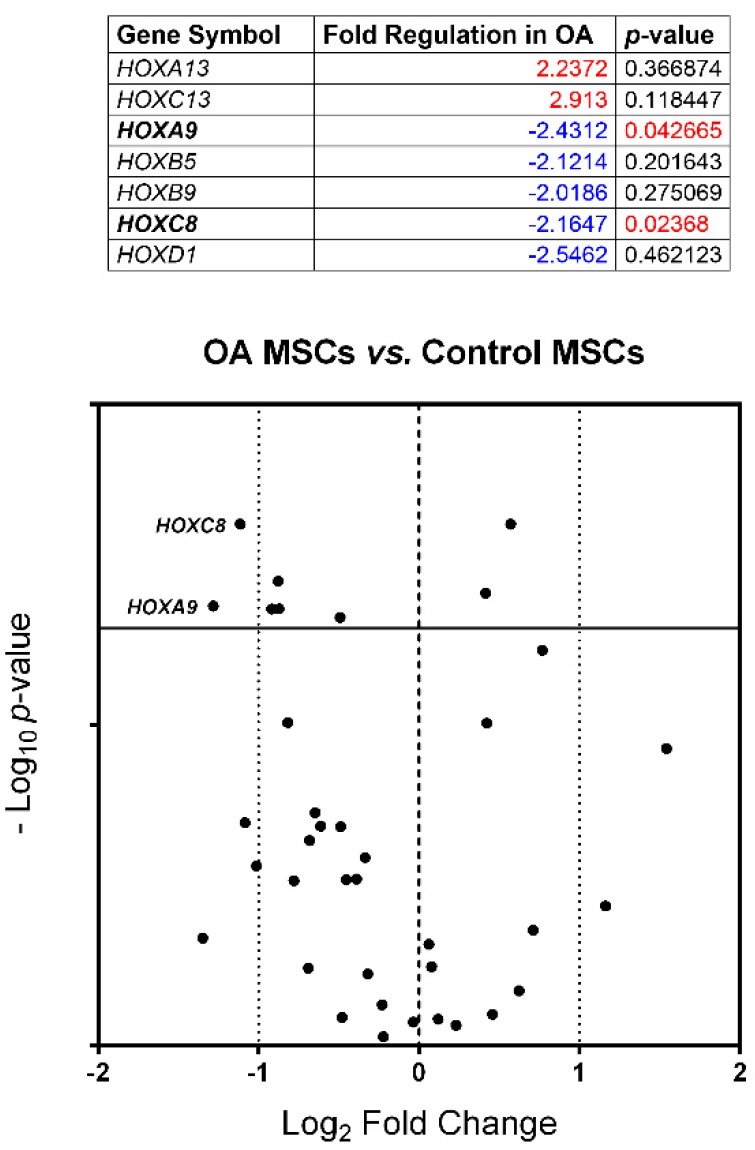
Numerical results and Volcano plot representation of *HOX* genes global expression. Genes upregulated and downregulated (negative values) in OA-Mesenchymal stem cells (MSCs) compared to Control-MSCs with fold differences greater than 2; *p*-values below 0.05 were considered statistically significant. The dashed lines indicate the expression boundaries. The solid lines indicate the 0.05 *p*-value threshold.

**Figure 3 cells-07-00244-f003:**
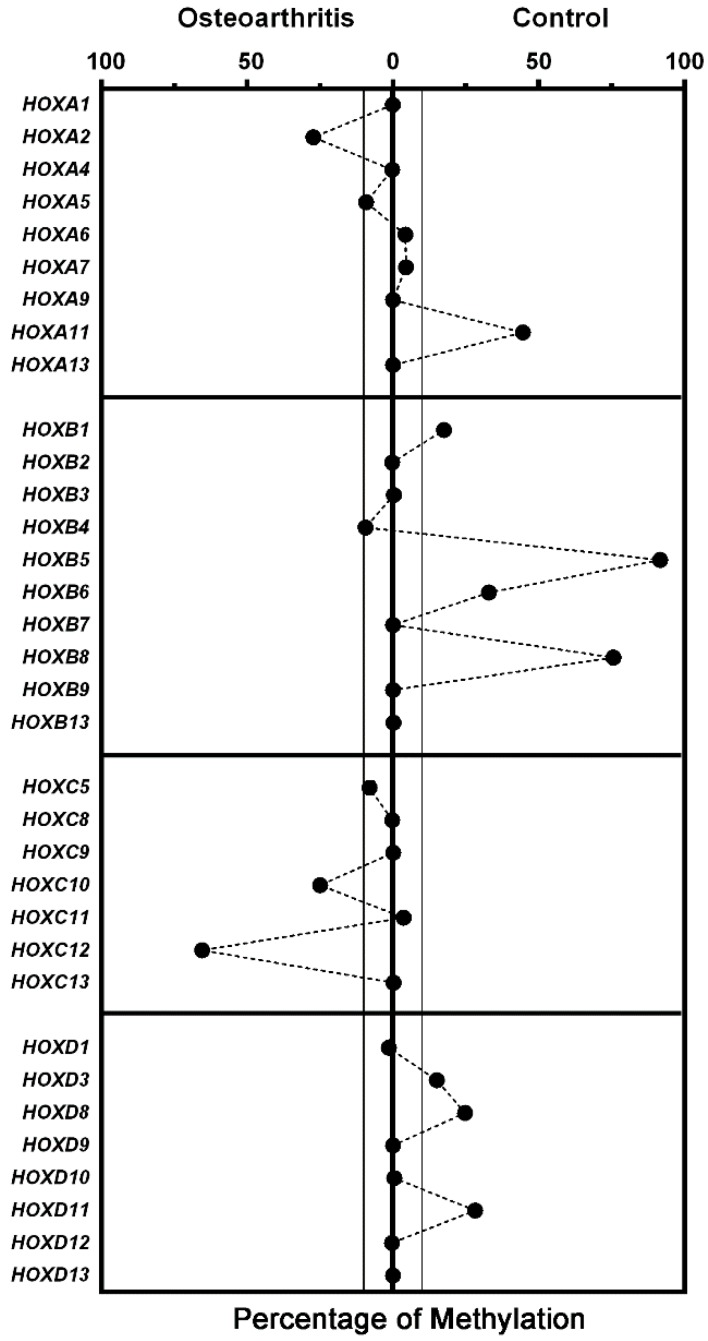
Methylation status of *HOX* gene promoters. The graph shows the difference in the percentage of methylation (hypermethylation) between OA-MSCs and Control-MSCs, calculated for each of the promoters of the indicated genes, grouped by clusters. The boundaries indicate methylation differences greater than 10%. Dashed lines between values have been drawn for ease of reading and data visualization.

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
