# Peer review of "Differential Expression of *HOX* Genes in Mesenchymal Stem Cells from Osteoarthritic Patients Is Independent of Their Promoter Methylation"

_cells, 2018, doi:10.3390/cells7120244_

Round 1
Reviewer 1 Report
The quality of manuscript has improved. However, the number of patients is too low. The manuscript should be improved by performing the miRNAs expression analysis.
Minor
It is not clear if the authors obtained the results from pooled
cells or individual patients cells (in this case the percentage media value
and SD of flow citometry data need to be showed).
The microscopy images should show a scale bar
Please specify the amount of bone aspirates from each patients
Author Response
Reviewer 1
The quality of manuscript has improved. However, the number of patients is too low. The manuscript should be improved by performing the miRNAs expression analysis.
We agree with the reviewer that a larger number would be more appropriate, but the biological material and analysis process are finite. Obtaining appropriate and correctly matched patients is a really complicated task. In addition, reviewers must consider that the initial microarray was performed using MSCs from different patients. At present it is impossible to increase the “N” because we are using new samples for other studies conducted in our laboratory. In any case, we consider that the results are indicative that among the possible epigenetic mechanisms involved, methylation of HOX gene promoters is not the preeminent.
In the discussion section, miRNAs have been mentioned as one of these other epigenetic alternatives but our intention was not to focus into it. Considering the inadequacy of pointing to the role of miRNAs without reliable evidence, we have decided to eliminate in the discussion any reference to them or other epigenetic regulatory mechanisms.
Minor
It is not clear if the authors obtained the results from pooled cells or individual patients cells (in this case the percentage media value and SD of flow citometry data need to be showed).
Results were obtained from individual patients. Changes required have been made in the corresponding image inside Supplementary File Figure S4
The microscopy images should show a scale bar.
These changes have been made in the corresponding image inside Supplementary File Figure S4
Please specify the amount of bone aspirates from each patients.
Each aspirate was performed during surgery for joint replacement.Thus, the number of aspirates as well as the number of patients is the same, one aspirate per patient.
Reviewer 2 Report
The
authors investigate the expression levels and the promoter methylation
status of a panel of HOX genes in MSCs obtained form OA or healthy
individuals. I
am convinced that informations provided in this manuscript are relevand
and of interest for the scientific community. Despite that, I do not
recommend acceptance of this manuscript in its current form due to
presence of a variety of formal shortcomings. These include mainly (but
not exclusively): typos & language -L113 …. The d ata were… -L171 … inespecific…. -L205 … depicted in figure 2, …. later on (e.g. L236) Figure 3 is used (figure vs. Figure) -L206 that (HOXA2, …. redundant parenthesis -L225, L227,L228 … pvalues…, whereas use of standard form in L241, L242 …p-value(s)… discrepances in the text -number
of HOX genes in analysis – 38 as mentioned in Abstract (L21) and
Materials and Methods (L161/162), but 39 genes mentioned in Results
(L204) and Discussion (L279) -use of wording Healthy-MSCs (L173) vs. Control-MSCs (L196); use of Control-MSCs vs. Control MSCs – use uniform style for the whole manuscript - others -L188 - L190 oddments from „Instructions to authors“ from the draft version??? Or is there any idea behind all this??? If so, it has to be clearly specified -L302 …, postraductional histone modifications …. ??? specify it
Author Response
The authors investigate the expression levels and the promoter methylation status of a panel of HOX genes in MSCs obtained form OA or healthy individuals.
I am convinced that informations provided in this manuscript are relevand and of interest for the scientific community. Despite that, I do not recommend acceptance of this manuscript in its current form due to presence of a variety of formal shortcomings. These include mainly (but not exclusively):
typos & language
Note that after extensive changes, number lines are different, but changes have been done
• -L113 …. The d ata were… (the space has been removed)
• -L171 … inespecific…. (It has been replaced by nonspecific L-159)
• -L205 … depicted in figure 2, …. later on (e.g. L236) Figure 3 is used (figure vs. Figure)
• In all cases figure was substituted by Figure
• -L206 that (HOXA2, …. redundant parenthesis (parenthesis has been removed)
• -L225, L227,L228 … pvalues…, whereas use of standard form in L241, L242 …p-value(s)…
In all cases pvalue was substituted by p-value
discrepances in the text
• number of HOX genes in analysis – 38 as mentioned in Abstract (L21) and Materials and Methods (L161/162), but 39 genes mentioned in Results (L204) and Discussion (L279)
That’s true, analysis was performed in 39 genes, however PCR was done only for 38. As described in supplementary file 1; according to Applied Biosystems re-evaluation, HOXC12 TaqMan® assay may detect genomic DNA and thus, It was not analyzed.
• use of wording Healthy-MSCs (L173) vs. Control-MSCs (L196); use of Control-MSCs vs. Control MSCs – use uniform style for the whole manuscrip.
Healthy-MSCs was changed by Control-MSCs. L160,184, 217, 222, 230, 236, 240, 284.
Others
• L188 - L190 oddments from „Instructions to authors“ from the draft version??? Or is there any idea behind all this??? If so, it has to be clearly specified.
I apologize for this mistake after using the template provided by MDPI, of course it was deleted from the manuscript.
• L302 …, postraductional histone modifications …. ??? specify it
L277. Covalent post-translational modification (PTM) to histone proteins includes methylation, phosphorylation, acetylation, ubiquitylation, and sumoylation, and can impact gene expression by altering chromatin structure or recruiting histone modifiers. As Histone proteins participate in DNA package, their chemical modifications can induce secondarily alterations in the transcriptional activation/inactivation.
Reviewer 3 Report
The authors have tried to link the HOX expression to OA. although, evidence exists that HOX may be involved in OA, in addition to the author’s gene expression analysis, the authors have not proven the influence of HOX on OA in this article. Epigenetic regulation (methylation) doesn’t seem to regulate the two significant HOX genes down regulated in the study, but, the authors suggest a role for miRNA in the regulation of those genes. They could have studied the expression of those miRNA in their OA samples. Overall the article doesn't address the OA and HOX connection but rather shows the HOX genes's expression are independent of methylation regulation.
Line 176 “Analysis of DNA microarray data revealed the lower expression of HOX genes in OA-MSCs” is confusing the authors should have used RNA as their starting material to study the expression profile and not DNA. Please change the heading to either expression analysis or something appropriate.
Line 211 – should not be DNA arrays rather expression arrays
Line 215 – HOXB8 is not present in the Figure 2 rather ist should be HOXC8
Line 181 – 182 : use fold change and upregulated and down regulated instead of over expressed and under expressed.
Line 191 change significative to significant
Please check for other typos
Author Response
Reviewer 3
The authors have tried to link the HOX expression to OA. although, evidence exists that HOX may be involved in OA, in addition to the author’s gene expression analysis, the authors have not proven the influence of HOX on OA in this article. Epigenetic regulation (methylation) doesn’t seem to regulate the two significant HOX genes down regulated in the study, but, the authors suggest a role for miRNA in the regulation of those genes. They could have studied the expression of those miRNA in their OA samples. Overall the article doesn't address the OA and HOX connection but rather shows the HOX genes's expression are independent of methylation regulation.
The evidences involving HOX expression anomalies in OA were also described in the microarray-based gene expression study. In this manuscript we only described that differential expression of HOX genes is independent of their promoter methylation. In agreement with the Reviewer the influence of HOX in OA was not proven but the existence of gene expression alterations, caused by any mechanism, is of sufficient importance to be studied. Probably the role of HOX in OA has a greater importance than that detected in our study, in which MSCs of OA origin have been used, but cultured under stable conditions without the pathological stress of origin.
In the discussion section, miRNAs have been mentioned as one of these other epigenetic alternatives but our intention was not to focus into it. Considering the inadequacy of pointing to the role of miRNAs without reliable evidence, we have decided to eliminate in the discussion any reference to them or other epigenetic regulatory mechanisms.
Line 176 “Analysis of DNA microarray data revealed the lower expression of HOX genes in OA-MSCs” is confusing the authors should have used RNA as their starting material to study the expression profile and not DNA. Please change the heading to either expression analysis or something appropriate.
I agree with the reviewer that it may be confusing, the correct name is: Microarray-Based Gene Expression Analysis, where purified RNA is used as template.
In the revised manuscript, the heading has been changed for the more appropriate:
“Microarray-Based Gene Expression Analysis reveals the lower expression of HOX genes in OA-MSCs”
Line 211 – should not be DNA arrays rather expression arrays
These changes have been made in the revised manuscript. . Note that line numbers might not match.
Line 215 – HOXB8 is not present in the Figure 2 rather ist should be HOXC8
These changes have been made in the revised manuscript. . Note that line numbers might not match.
Line 181 – 182 : use fold change and upregulated and down regulated instead of over expressed and under expressed.
These changes have been made in the revised manuscript. Note that line numbers might not match.
Line 191 change significative to significant
These changes have been made in the revised manuscript
Please check for other typos
Typos checked, as much as possible.
Round 2
Reviewer 1 Report
The authors replied satisfactorily
Reviewer 2 Report
All points were answered, I do not have any further comments/improvement suggestions. The revised manuscript can be published in its current form.
Reviewer 3 Report
The authors have addressed the concerns satisfactorily